# Purified Native and Recombinant Major *Alternaria alternata* Allergen (Alt a 1) Induces Allergic Asthma in the Murine Model

**DOI:** 10.3390/jof7110896

**Published:** 2021-10-24

**Authors:** Ainara Vélez-del-Burgo, Patricia Sánchez, Ester Suñen, Jorge Martínez, Idoia Postigo

**Affiliations:** Department of Immunology, Microbiology and Parasitology, Faculty of Pharmacy and Laboratory of Parasitology and Allergy, Lascaray Research Centre, University of the Basque Country, 01006 Vitoria-Gasteiz, Spain; ainara.velezdelburgo@ehu.eus (A.V.-d.-B.); patricia.sanchez@ehu.eus (P.S.); ester.sunen@ehu.eus (E.S.); jorge.martinez@ehu.eus (J.M.)

**Keywords:** asthma, *Alternaria alternata*, Alt a 1, mice

## Abstract

Aeroallergens such us the spores of *Alternaria alternata* are described as the most important agents associated with respiratory allergies and severe asthma. Various experimental models of asthma have been developed using *A. alternata* extracts to study the pathogenesis of asthma, establishing the main parameters that trigger the asthmatic response. In this study, we describe a mouse model of asthma induced only by Alt a 1. To induce the allergic response, mice were challenged intranasally with the major allergen of *A. alternata*, Alt a 1. The presence of eosinophils in the lungs, elevated concentrations of Th2 family cytokines, lymphocyte proliferation and elevated IgE total serum levels indicated that the sensitisation and challenge with Alt a 1 induced the development of airway inflammation. Histological studies showed an eosinophilic cellular infiltrate in the lung tissue of mice instilled with Alt a 1. We demonstrate that Alt a 1 alone is capable of inducing a lung inflammatory response with an increase in IgE serum levels mimicking the allergic asthma immunoresponse when it is administered into BALB/c mice. This model will allow the evaluation of the immunoregulatory or immunotolerant capacity of several molecules that can be used in targeted immunotherapy for fungal allergic asthma.

## 1. Introduction

Asthma is a complex inflammatory disease clinically characterised by airway hyperresponsiveness, inflammatory cell infiltration in the bronchoalveolar lavage (BAL) and bronchial walls and structural changes to the airway [1]. Asthma affects more than 300 million people worldwide [2] and type 2 allergic asthma is one of the most widespread phenotypes of the disease [3]. The entry and recognition of allergens by the antigen-presenting cells (APC) triggers a series of immune responses in which Th2 cells and innate lymphoid cells type 2 (ILC2) play a major role in the development of inflammation and allergic response [4]. Th2 cells participate in the adaptive immune response, producing Th2-type cytokines such as IL-4, IL-5 and IL-13, activating macrophages, producing specific or total IgE and recruiting eosinophils [5]. On the other hand, ILC2s present in epithelial cells are activated by IL-25, IL-33 or TSPL cytokines called “alarmins” which are local immune mediators in the innate immune response. ILC2, such as Th2 cells, express IL-5 and IL-13, but very low levels of IL-4 [6]. These type 2 cytokines cause the recruitment and activation of eosinophils, mast cells and basophils. Therefore, both ILC2 and Th2 cells are responsible for triggering type 2 allergic asthma [7].

Several animal models are described that reproduce the classical parameters of asthma [8]. To study allergic asthma, the most commonly used species in the experimental models is the mouse [9], in which the BALB/c strain is one of the most relevant [10]. These mice, along with the A/J and C57BL/6 strains, develop a Th2-type immune response when asthmatic disease is artificially induced by the sensitisation to the allergen and then by the subsequent challenge of the allergen [11,12]. BALB/c and C57BL/6 mice exhibit different degrees of airway response and inflammation despite being sensitised and administered with the same allergen [13]. BALB/c mice are more predisposed to develop elevated levels of IgE and IgG1; elevated levels of Th2 cytokines such as IL-4, IL-5 and IL-13; and increased inflammation and bronchial hyperresponsiveness, compared to C57BL/6 mice [14]. However, C57BL/6 mice present a higher number of eosinophils in the airways [15]. It is clear that the genetic background of mice has a great impact on the development of asthma [16], as C57BL/6 mice show less allergic symptomatology than BALB/c mice in all the characteristics evaluated, except for eosinophilia [17].

The most common route of sensitisation is the intraperitoneal administration of the allergen, although subcutaneous administration is also possible [18]. In this way, one or two intraperitoneal injections are usually necessary on days 0 and 7, followed by a subsequent challenge with the allergen, which may require several days [19]. To increase the immunogenicity of the allergens, adjuvants such as aluminium sulphate can be used [20]. However, several studies claim that these methods of sensitisation do not accurately mimic the natural sensitisation process in humans and propose that sensitisation must be induced by inhalation [21]. Allergen challenges to produce an asthmatic response can be carried out intratracheally [22], intranasally [23] or by aerosols [24], but recent studies demonstrate that the best route of the challenge is the inhaled route, specifically the intranasal route, as the allergen gains direct and rapid access to the airways, compared to the intratracheal route [25].

Fungal allergens are among the most relevant allergens for inducing asthma, as the abundant presence of spores and conidia in both indoor and outdoor environments increases the exposure to these aeroallergens, as opposed to pollen [26], house dust mites and cockroaches [27]. The most studied allergenic fungi are *Aspergillus, Candida, Penicillium, Alternaria and Cladosporium* spp. [28]. *Alternaria* and *Cladosporium* spp. by releasing their allergens into the environment can cause rhinitis, atopy and asthma exacerbations, but rarely cause infections [29]. *Aspergillus* spp., aside from these diseases, can also cause infections, such as allergic bronchopulmonary aspergillosis (ABPA) [30]. Forkel et al., studied the prevalence of sensitisation to these three fungi in Europe over 20 years (1998–2017). In the first decade, 19.2% of patients tested positive for at least one of them (*Alternaria, Aspergillus or Cladosporium*), while in the second decade, this figure increased to 22.5%. In this last stage, they observed an increase in the number of polysensitised patients, indicating the relevance of these moulds in the allergic disease [31].

Among fungi, one of the most allergenic fungal species is *A. alternata* [32,33]. This mould is considered to be an important factor in the development and exacerbation of asthma, especially in children [34]. *Alternaria* species possess metabolic and mycelial antigens capable of causing allergic responses [28]. Fifteen allergens of *A. alternata* have been described to date [35] and, Alt a 1, the major allergen of *A. alternata*, is considered to be the major marker of sensitisation as it is recognised by 80% of patients allergic to this mould [36,37]. Alt a 1 is found in the cell wall of spores and can also be found as a predominant component in mycelial extracts [38]. Leino et al. and Causton et al. used *A. alternata* extract to produce allergic asthma in mice because its antigenic components, such as chitin and/or proteases, contributed to the development of murine models of allergic asthma [39,40]. *A. alternata* serine protease present in the fungal extract induces changes in the respiratory epithelial barrier leading to alterations in cell–cell junctions and the reorganisation of the actin cytoskeleton [41]. However, Garrido-Arandia et al. [42] described an in vitro cellular model of asthma where Alt a 1 was able to interact with Calu-3 airway epithelial cells, inducing the proliferation of inflammatory cytokines such as IL-8 and the cytokines IL-25 and IL-33. These authors hypothesised that Alt a 1 may mimic human siderocalin as a ligand-free route of entry into the airway mucosa. These authors also highlighted that sensitivity to Alt a 1 was a risk factor for developing severe and life-threatening asthma [43].

In this study, we established a murine model of fungal allergen-induced asthma in which Alt a 1 alone was responsible for inducing the classic parameters of the allergic disease.

## 2. Materials and Methods

### 2.1. Alternaria alternata Extract and Purification of Native and Recombinant Alt a 1

Extracts of *A. alternata* were obtained from *Centraal Bureau voor Schimmelcultures’* (CBS) strain 103.33. This mould was grown at 24 °C on Sabouraud agar plates for 4 weeks. Mould cultures were homogenised with phosphate saline buffer (PBS) and stirred at 4 °C overnight. After this, they were centrifuged for 20 min at 10,000 rpm and sterilised through 22 µm filters. Finally, they were dialysed against distilled water and lyophilised for storage at 4 °C [44].

Native Alt a 1 (nAlt a 1) was purified from frozen-dried extracts of *A. alternata* by affinity chromatography using Dynabeads (Invitrogen, Waltham, MA, USA) cross-linked with rabbit anti-Alt a 1 IgG. After 30 min of room temperature incubation (extracts with the Dynabeads) and several washes with PBS, Alt a 1 was finally eluted by adding Glycine 50 mM, pH 2.8 [45].

Recombinant Alt a 1 (rAlt a 1) was cloned into the expression vector pET28a and trans-transformed into E. coli C43 (DE3) pLyss strain (Lucigen, Middleton, WI, USA). The rAlt a 1 was purified by affinity chromatography using HisPur Ni-NTA spin columns (Thermo Fisher, Waltham, MA, USA). Amicon Ultra Centrifugal Filters MWCO 10 kDA (Sigma-Aldrich, St. Louis, MO, USA) were used to concentrate the protein [45].

### 2.2. SDS-PAGE Immunoblotting

Electrophoresis of both native and recombinant Alt a 1 proteins from *A. alternata* was performed by SDS-PAGE, using 12.5% polyacrylamide separating gels [46]. Separated proteins were electrophoretically transferred to PVDF membranes (Immobilon-P, Millipore, Burlington, MA, USA), according to Saénz de Santamaría et al. [47]. The membrane containing nAlt a 1 was incubated with *A. alternata* allergic patients’ sera (sIgE > 5 kU/L. ImmunoCap^®®^) by gentle shaking for 18 h at 4 °C. Finally, 1:80,000 anti-IgE human secondary antibody (Thermo Fisher) was used to detect nAlt a 1. rAlt a 1 was detected using an 1:10,000 anti-His antibody (Thermo Fisher). This development was made by chemiluminescence (Thermo Fisher) using a ChemiDoc image analysis system (BioRAD, Hercules, CA, USA).

The rAlt a 1 protein was cut from the SDS-PAGE gel, digested and cleaned for analysis by liquid chromatography and mass spectrometry.

### 2.3. Endotoxin

Endotoxin or LPS is present in the membrane of Gram-negative bacteria and causes inflammation and toxicity in humans [48]. The presence of endotoxin in purified Alt a 1 samples was tested using the Pierce Chromogenic Endotoxin Quantitation Kit (Thermo Fisher) following the manufacturer’s instructions. Briefly, endotoxin standards, blanks and samples were added to the pre-warmed plate at 37 °C. Then, 50 µL of amebocyte lysate was added and incubated for 1 h, after which 100 µL of chromogenic substrate was poured. Six minutes later, the stop solution was added, and the OD was read at 405 nm.

### 2.4. Mice

Forty female BALB/c mice between six and eight weeks old were obtained from Janvier Labs and housed under controlled temperature, humidity, and day-night cycles. The experiments were approved by the Animal Experimentation Ethics Committee of the University of the Basque Country UPV/EHU (license M20/2018/204) following Spanish and European Union (EU) laws and all the procedures were followed in accordance.

Mice were assigned to the following groups: Control groups: 10 mice instilled with PBS; 10 mice instilled with *A. alternata* extract (*A. a.* extract); Treated groups: 10 mice instilled with nAlt a 1; 10 mice instilled with rAlt a 1.

Mice were anesthetised with isoflurane (3%) and instilled intranasally for four consecutive days with 20 µg of *A. alternata* extract, 5 µg of rAlt a 1 or 5 µg nAlt a 1, respectively, in a final volume of 20 µL PBS. PBS control group received 20 µL of PBS at all times. After 24 h of the last exposure, on the fifth day, mice were euthanised by cervical dislocation [49].

### 2.5. Analysis of the Bronchoalveolar Lavages (BALs)

Just after the mice were euthanised, a 1.2 mm needle was inserted through the exposed trachea and the bronchoalveolar lavage supernatants (BALs) were collected by washing the lungs three times with 1 mL of sterile PBS.

The BAL was centrifuged, and the cells were fixed in 1% of paraformaldehyde for 10 min. Then, the cells were washed with FACS (PBS, 4% FCS and 0.01% sodium azide). Then, a mix of antibodies (Alexa Fluor 488 conjugated CD45, PE conjugated Siglec-F, Efluor 450 conjugated CD11c, APC conjugated CD11b and Fc block) was added to mark the eosinophils [50].

### 2.6. Histological Analysis

The right lung was removed and stored in a 4% formalin solution. After overnight fixation, it was transferred to a sucrose solution for crystal removal for 24 h and after that, the lungs were embedded in resin (Tissue Tek, O.C.T. Sakura, Europe). Portions of 5 µm were cut with a cryostat (Cryocut 3000, Leica, Bensheim, Germany) and stained with a Hematoxylin Eosin (HE) solution [51].

### 2.7. Proliferation of Lymphocytes

Spleens were homogenised and splenocytes were seeded at a concentration of 5 × 10^5^ cells per well in cell culture medium. The cells were then stimulated with *A. alternata* extract or nAlt a 1 according to the group to which the animals belonged. Only cell culture medium was added to the splenocytes in the control group (instilled with PBS). After incubation for 3 days at 37 °C and 5% CO_2_, 50 µL of MTT was added to each well and incubated for four hours. The reading was performed after adding 200 µL of DMSO [52].

### 2.8. Cytokines and Total IgE Quantification by ELISA

Both left lung and spleen were removed. The measurement of the immune response was performed using the supernatant of the homogenised lung and spleen. These organs were analysed for IL-5, IL-13, IL-17E (IL-25) and IL-33 cytokines by ELISA DuoSet from R&D Systems (Minneapolis, MN, USA) following the manufacturer’s instructions.

Blood was collected by cardiac puncture and the total serum IgE was measured using the mouse IgE ELISA Kit (Invitrogen), according to the manufacturer’s instructions.

### 2.9. Statistical Analysis

Statistical analysis was performed using Prism software (version 7.0, GraphPad Software, Inc., La Jolla, CA, USA). All the results were analysed through one-way ANOVA and Bonferroni post hoc tests. The differences between the groups were considered significant when *p* values less than 0.05 were obtained.

## 3. Results

### 3.1. Alternaria alternata Extracts and Purification of Native and Recombinant Alt a 1

Native Alt a 1 (nAlt a 1) was obtained from *A. alternata* extracts produced in our laboratory, while recombinant Alt a 1 (rAlt a 1) was produced in E. coli C43 (DE3) pLyss strain (Lucigen).

Figure 1 shows the immunodetection of nAlt a 1 in *A. alternata* extract (lane 1) and nAlt a 1 following purification (lane 2) and rAlt a 1 immunodetection with an anti-His antibody (lane 3). In both the extract and nAlta 1, a dimeric protein of 14.5 kDa molecular weight was observed and, a 25 kDa molecular weight protein was detected in rAlt a 1. The results obtained by the mass spectrometry of the rAlt a 1 protein showed 98% homology with the native Alt a 1 sequence (NCBI. BBK07919.1).

### 3.2. Quantification of Total IgE in Mouse Serum

Figure 2 shows that all of the groups instilled with *A. alternata* extract or Alt a 1 (native and recombinant) achieved IgE levels with statistically significant differences (*p* < 0.001) compared to the mice administered with PBS. The highest IgE levels were obtained by the group of mice instilled with nAlt a 1 compared to control groups and rAlt a 1-instilled mice (*p* < 0.001). No statistically significant differences (*p* > 0.05) were demonstrated between the groups instilled with *A. alternata* extract and the group of mice instilled with rAlt a 1.

### 3.3. Cytokine Quantification in Lung Homogenate

The instillation of nAlt a 1 resulted in an increase in IL-5, IL-13, IL-17E and IL-33 in lung homogenates compared to the PBS control group (*p* < 0.001) (Figure 3). Instillation of rAlt a 1 induced the IL-5, IL-13 and IL-33 cytokines production (*p* < 0.001) but, the IL-17E levels induced by rAlt a 1 demonstrated no statistically significant differences (*p* > 0.05) compared with the control groups. An increase in IL-17E and IL-5 was observed in the groups of animals instilled with *A. alternata* extract compared to mice instilled with PBS (*p* < 0.01 and *p* < 0.05, respectively). The IL-13 and IL-33 cytokine levels induced by nAlt a 1 and rAlt a 1 instillation were higher than those induced by *A. alternata* extract (*p* < 0.001). No statistically significant differences (*p* > 0.05) in IL-13 and IL-33 levels were demonstrated between *A. alternata* extract and PBS-instilled mice.

### 3.4. Cytokine Quantification in Spleen Homogenate

Figure 4 shows that the instillation of nAlt a 1 induced the IL-5, IL-17E and IL-33 cytokines production (*p* < 0.001) but, no statistically significant differences (*p* > 0.05) were demonstrated in IL-13 levels produced by nAlt a 1 instilled mice compared to the PBS control group. The instillation of rAlt a 1 resulted in an increase in IL-5, IL-13, IL-17E and IL-33 levels in spleen homogenates compared to the PBS control group (*p* < 0.05 and *p* < 0.001). The IL-13 and IL-17E levels were increased (*p* < 0.001) in mice instilled with *A. alternata* extract compared to the PBS control group but, no statistically significant differences (*p* > 0.05) were observed in the IL-17E levels between the treated groups and the mice instilled with *A. alternata* extract. No statistically significant differences (*p* > 0.05) were demonstrated in the production of IL-5 and IL-33 cytokines between the control groups.

The IL-33 cytokine levels induced by the nAlt a 1- or rAlt a 1-treated groups were higher than those induced by the *A. alternata* extract (*p* < 0.001 and *p* < 0.05). This control group demonstrated no statistically significant differences (*p* > 0.05) compared to the PBS-instilled mice.

### 3.5. Analysis of Lymphocyte Proliferation

Figure 5 shows that *A. alternata* extract and nAlt a 1 were able to induce lymphocyte proliferation compared to the control group (PBS). These results showed statistically significant differences (*p* < 0.001). The *A. alternata* extract-instilled mice showed the highest lymphocyte proliferation rate. Statistically significant differences (*p* < 0.001) between the animals instilled with the *A. alternata* extract and the animals instilled with nAlt a 1 were detected. Lymphocyte proliferation was not measured in mice instilled with rAlt a 1.

### 3.6. Eosinophil Count

Figure 6 shows the results obtained for eosinophil counts. A BAL analysis confirmed statistically significant differences (*p* < 0.001) in lung eosinophil counts between treated groups (mice instilled with nAlt a 1 or rAlt a 1) and control groups (mice administered with PBS or *A. alternata* extract). The highest number of eosinophils observed was in mice instilled with nAlt a 1. No statistically significant differences (*p* > 0.05) were found between the treated groups (mice instilled with nAlt a 1 or rAlt a 1) or between the control groups (mice instilled with PBS or *A. alternata* extract).

### 3.7. Histopathological Analysis of Lung Tissue

The examination of the lung tissues confirmed an eosinophilic cellular infiltrate in those groups of mice instilled with the *A. alternata* extract or its major allergen, either in its native or recombinant form. However, this infiltrate was more noticeable in the group of animals instilled with nAlt a 1. Peribronchial thickening was also observed in all groups except the PBS group (Figure 7).

## 4. Discussion

In this study, we established a murine model of allergic asthma that reproduced the fundamental parameters of asthmatic disease, using the major allergen of *A. alternata*, and Alt a 1 as the only inducer.

To establish the murine model we selected 6–8 week old female BALB/c mice, as this strain, compared to C57BL/6 mice, best reproduced allergic asthma [13,14,15,16,17,53]. Young mice were selected because cellular immunity decreased as the individuals aged [54,55]. Additionally, all mice were female as several studies claimed that they were more susceptible to developing stronger immune responses than males [56,57].

The most widely used model to produce allergic asthma in mice is the model that uses ovalbumin (OVA) as an allergen [58]. However, OVA is not an aeroallergen and the prior sensitisation by the intraperitoneal or inhalation administration of this egg protein, followed by subsequent challenges, is required to establish the asthma model [59]. It has been shown that the prolonged exposure for several weeks to OVA does not lead to chronic airway inflammation, but results in tolerance and a decreased inflammation [60,61]. Therefore, the use of aeroallergens that are more representative of allergic disease and only need to be inoculated through the airways over several days, such as pollen [62], house dust mites [63] or fungal allergens [64], is becoming more widespread.

In our study, we intranasally administered mice for four consecutive days with *A. alternata* extract and its major allergen, Alt a 1. We showed that Alt a 1, either native or recombinant, increased the total IgE values, irrespective of the absence of prior intraperitoneal sensitisation. This result is comparable with the results of Li-Ping Thio et al. and Cates et al., showing that, with a more physiological sensitisation through intranasal administration, it is possible to induce the IgE response [49,65]. Thus, our model replicated the natural process of allergen exposure in asthmatic patients. The instillation with Alt a 1 caused an increase in the levels of the so-called “alarmin” cytokines, IL-17E (IL-25) and IL-33, similar to that observed in the model of Yi et al. when administering OVA intranasally [66]. The cytokines, IL-17E and IL-33, induced the release and proliferation of the innate lymphoid cell type 2 (ILC2), which in turn secreted Th2 cytokines (IL-4, IL-5 and IL-13) [67]. Our model with nAlt a 1 and rAlt a 1 induced the elevated production of the cytokines IL-5 and IL-13, resulting in statistically significant differences (*p* > 0.001) compared to the control groups. These results are consistent with those presented by Gil et al. [68], who obtained cytokine levels of IL-5 and IL-13 close to 1000 pg/mL, 24 h after the last administration of the *A. alternata* extract. In contrast to some models that measured IL-4 [69], in our model we measured the Th2 response through IL-5 and IL-13, because they were able to envelop the parameters of asthma [70]. Recent studies suggest that eosinophil-derived IL-13 may play a more prominent role than IL-4 in the establishment of allergic inflammation in mice models of asthma [71]. It is also shown that the cytokines IL-4 and IL-13 share a common receptor subunit and that both interleukins can induce IgE production or regulate macrophage activity [72].

In our model, after four days of intranasal challenges, the Th2 cytokine levels in the spleen were comparable to those in the lung. This showed that there was a migration of dendritic cells from the lung to the lymphoid organ that led to a systemic Th2-type response. After the stimulation of splenocytes, lymphocyte proliferation was observed in mice that had been instilled with fungal extract and native Alt a 1. Lymphocyte proliferation is a key feature of the lymphocyte response to antigenic stimulation, leading to an effective immune response to antigen [73].

In addition to the total IgE and Th2 cytokines, we measured eosinophilia in the lungs of mice, an important parameter that allowed us to distinguish between the different asthma phenotypes [74]. We observed increased numbers of eosinophils in the bronchoalveolar lavage and increased epithelial thickening and eosinophilic infiltrates in the histological sections of mice instilled with the Alt a 1, confirming lung inflammation. The release of proinflammatory mediators by eosinophils is the main cause of inflammation in allergic asthma [75]. Eosinophils release toxic granule proteins, reactive oxygen species (ROS) and cytokines such as IL-5, which cause airway epithelial cell damage, mucus hypersecretion and airway remodeling [76]. Our results were comparable to those presented in several asthma models, where the number of eosinophils detected in BAL exceeded the 3% of eosinophils that could be found in BAL of asthmatic patients [11]. Since the LPS concentrations detected in solutions containing Alt a 1 were 0.003 EU/mL, it was highly unlikely that the elevated eosinophil numbers were caused by endotoxins. As demonstrated by Kouzaki et al., it takes more than 1 µg of LPS to produce eosinophilia [77].

We observed differences in Th2 cytokine behaviour among mice instilled with *A. alternata* extract and the mice administered with Alt a 1. Mice that received the fungal extract showed elevated levels of the cytokines IL-5 and IL-17E, but not of the cytokines IL-13 and IL-33. Most authors reported an increase in IL-33 levels in BAL one to six hours after the last instillation [78], then the levels of IL-33 in BAL tended to normalise at 24 h [79]. This may be because the release of IL-33 preceded the releases of IL-5 and IL-13, resulting in low levels 24 h after the last exposure to the fungal extract [80]. The cytokine IL-17E can act independently of IL-33 in the production of the allergic response [81]. Terrier et al. demonstrated that IL-17E could induce the release of IL-5 through mononuclear cells and cause eosinophil recruitment [82]. Both in mice instilled with *A. alternata* extract and in those instilled with Alt a 1, it could be observed that the cytokines detected (IL-17E or IL-33) were the “alarmin” cytokines that activated ILC2 in the innate immune response.

This variation in results between the groups receiving *A. alternata* extract or its main allergens may also be due to the difference in the compositions of the preparations. In the extract, in addition to Alt a 1, other proteins were found, such as proteases and chitins [28]. These proteases were able to induce changes in the epithelial barrier, allowing the entry of *A. alternata* allergens, which could lead to a different behaviour of the immune system [41].

Another reason for this variation could be the different amount of Alt a 1 present in each preparation, being up to 10 times lower in the extract. The amount of fungal extract that needs to be instilled into the mice to balance the Alt a 1 levels of the nAlt a 1-instilled mice would be too high to guarantee the survival of the animals. The disparity between our results and those of the models using *A. alternata* extract suggest that not all extracts have the same allergen load. These variations can be avoided using inducers with a specific and measurable protein content, capable of producing allergic responses such as major allergens [83].

We demonstrated that Alt a 1, either in its native or recombinant form, was able to induce allergic responses in BALB/c mice. The animals developed allergic lung inflammations characterised by the local secretion of Th2 cytokines, high numbers of eosinophils in the airways and elevated IgE production. The establishment of this allergic asthma model induced by Alt a 1 is the most major cause of *A. alternata* fungal asthma and its exacerbation [36,43], and allows the evaluation of the immune-regulatory or immune-tolerant capacity of several molecules that could be used in targeted immunotherapy treatments for this growing disease.

## Figures and Tables

**Figure 1 jof-07-00896-f001:**
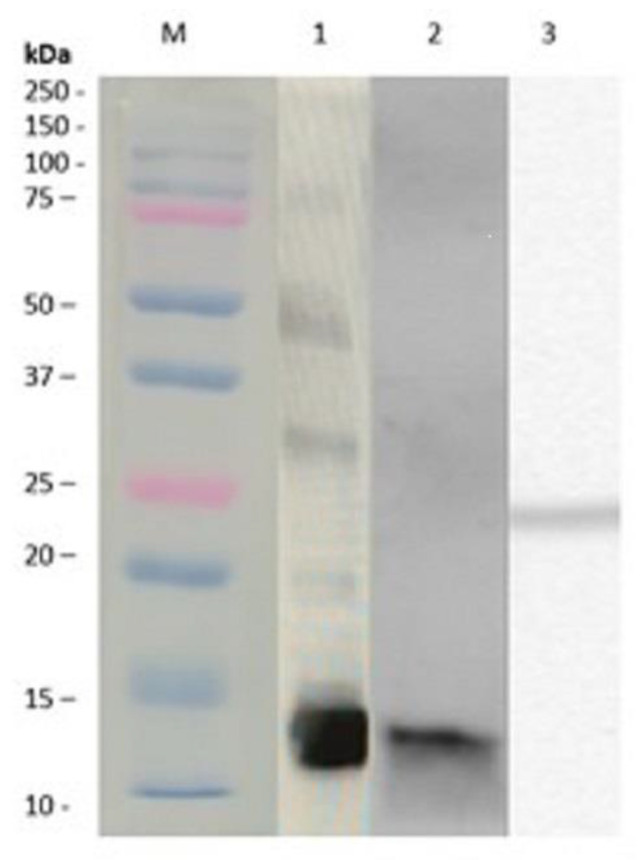
Identification of purified nAlt a 1 and rAlt a 1 by immunoblotting. Lane 1: *A. alternata* extract, Lane 2: purified nAlt a 1, Lane 3: purified rAlta 1. M: Molecular weight marker.

**Figure 2 jof-07-00896-f002:**
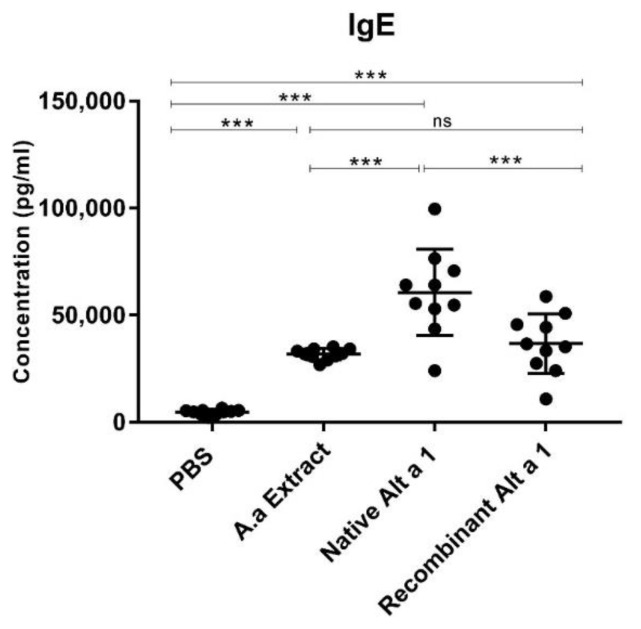
Quantification of total serum IgE in each group of BALB/c mice studied. Lines represent the mean and bars the SEM. ***, *p* < 0.001; ns, no statistically significant differences (*p* > 0.05).

**Figure 3 jof-07-00896-f003:**
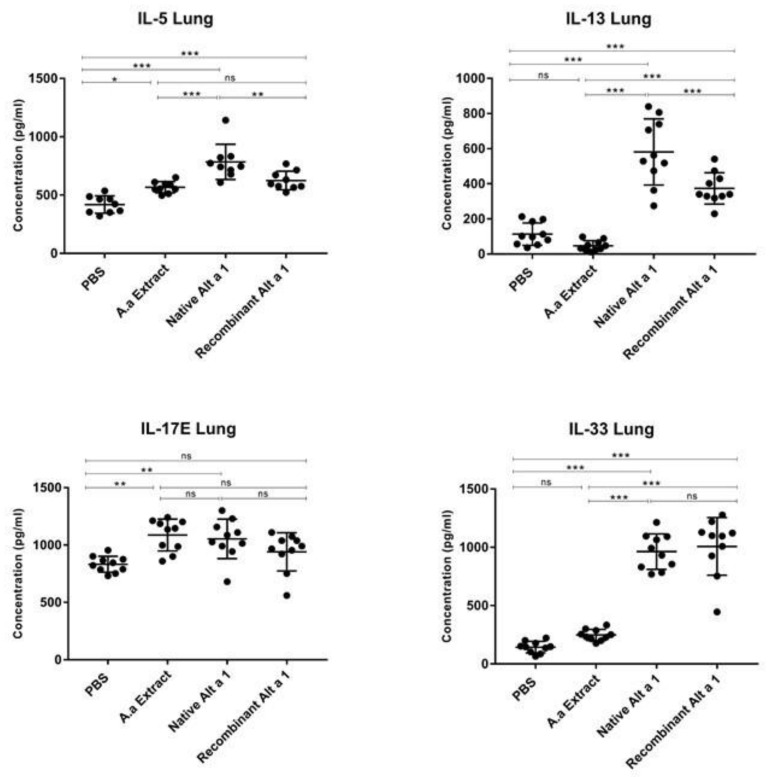
Quantification of IL-5, IL-13, IL-17E and IL-33 cytokines in lung homogenate in the groups of mice instilled with PBS; *A.alternata* Extract; native Alt a 1 (nAlt a 1) or with recombinant Alt a 1 (rAlt a 1). Lines represent the mean and bars the SEM. *, *p* < 0.05 **, *p* < 0.01; ***, *p* < 0.001; ns, no statistically significant differences (*p* > 0.05).

**Figure 4 jof-07-00896-f004:**
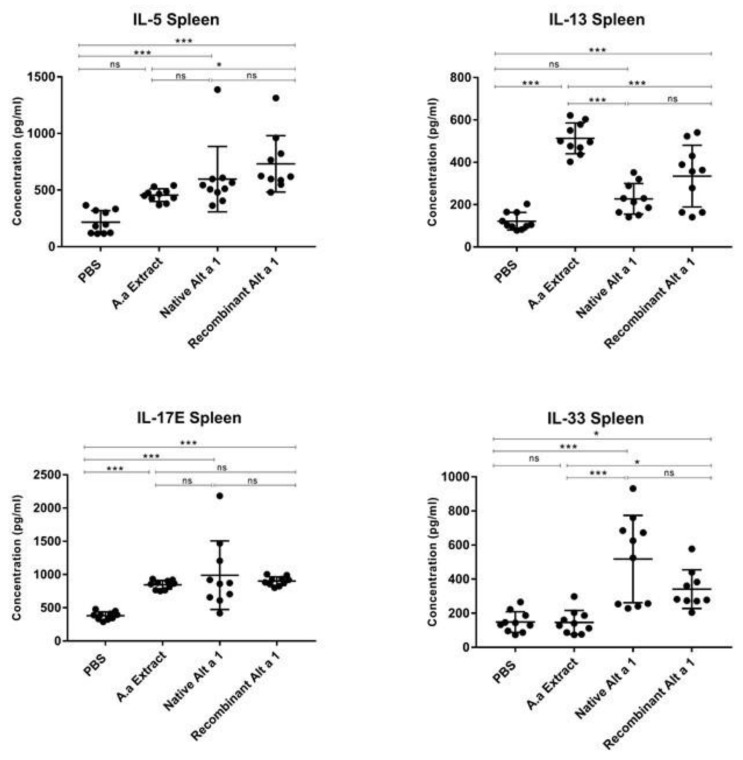
Quantification of the levels of cytokines, IL-5, IL-13, IL-17E and IL-33, analysed by ELISA in spleen homogenate in the respective groups of animals (PBS, *A. alternata* Extract, nAlt a 1 and rAlt a 1). Data shown as mean ± SEM. *, *p* < 0.05; ***, *p* < 0.001; ns: no statistically significant differences (*p* > 0.05).

**Figure 5 jof-07-00896-f005:**
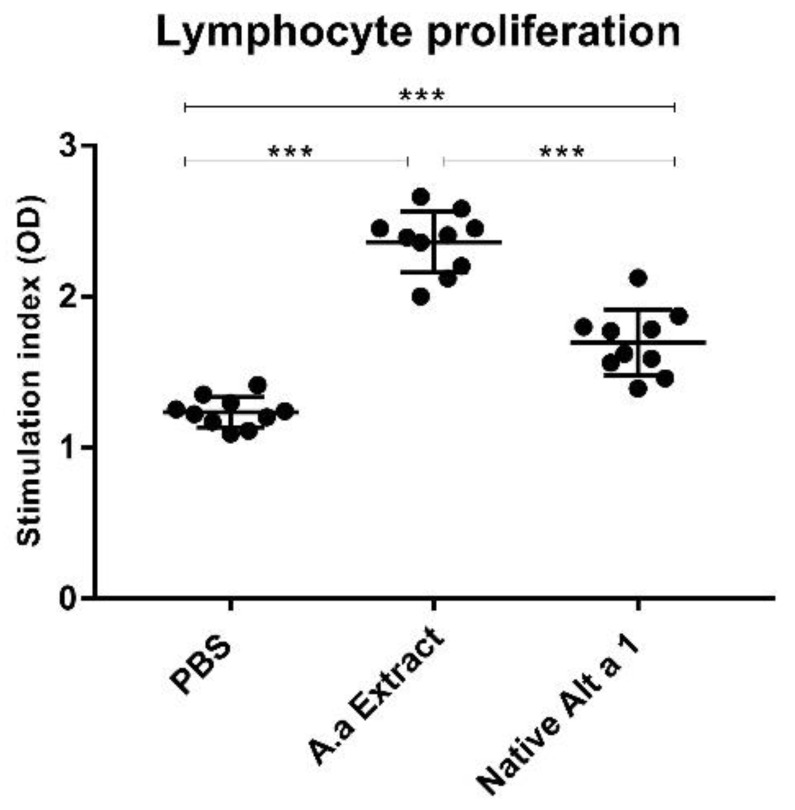
Stimulation index (OD) in lymphocyte proliferation. The OD was calculated as the ratio between the absorbance values of experimental (*A.alternata* Extract and nAlt a 1) and control samples (PBS). Significant differences (*** *p* < 0.001) were observed between the nAlt a 1, the *A. a* extract and the PBS-instilled mice.

**Figure 6 jof-07-00896-f006:**
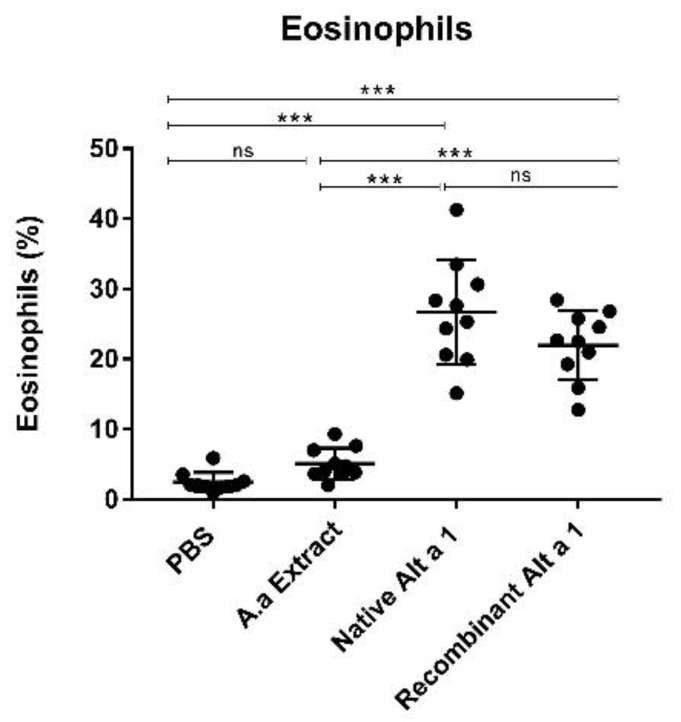
Quantification of eosinophil in BAL. Eosinophils are plotted as the percentage of the mean of the different groups (PBS, A. alternata Extract, nAlta 1 and rAlta 1). Data shown as mean ± SEM. ***, *p* < 0.001; ns: no statistically significant differences (*p* > 0.05).

**Figure 7 jof-07-00896-f007:**
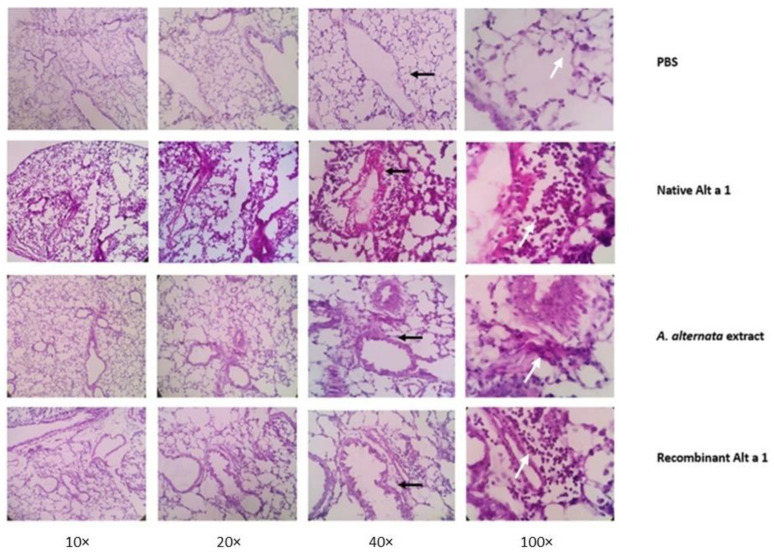
Images of lung histological sections performed on each group of animals (PBS, nAlt a 1, *A.alternata* Extract and rAlt a 1). Lungs were collected 24 h after the last instillation and formalin-fixed and stained with H&E. The black arrows point to the thickened peribronchial tissue in the lungs of the different groups of mice while the white arrows point to the eosinophils present in the epithelial lung tissue of the mice instilled with the control and Alt a 1 solutions.

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
