# Peer review of "Purified Native and Recombinant Major Alternaria alternata Allergen (Alt a 1) Induces Allergic Asthma in the Murine Model"

_jof, 2021, doi:10.3390/jof7110896_

Round 1
Reviewer 1 Report
In this article, authors develop a murine model of asthma induced by Alternaria alternata allergen. This work provides a novel and valuable information about this model, which can be used in future experiments to study immunoregulation and immunomodulation in asthma disease. Although a lot of aspects are clear, I have several concerns about it:
MAJOR COMMENTS
- Figure 2, 3 and 4: Comparisons between groups (Extract, nAlt a 1 and rAlt a 1) are missed. Are there statistical differences between these groups?
- Page 7, Lines 236-237: Why did you not measure lymphocyte proliferation in mice instilled with rAlt a 1?
- Page 6 and 7, Lines 243-255 (Flow cytometry results): In my opinion I believe that these experiments are not well conducted. First, you must show the gating strategy used (FSC vs SSC, single cells, CD45+ cells, SIGLEC-F+/CD11B+ cells, etc.). Second, in dot plots I do not distinguish independent populations and fluorescence intensity are very low; please, show me isotype controls that you use, and the gating strategy followed. How do you calculate the number eosinophils/mL, what software did you use? I think that the better data is eosinophil percentage. Also, you must add a graph where you show the percentage values of eosinophils from each mouse, and reflect on it the significant differences.
- Page 8, Line 262 (Figure 7): It would be interesting to include a graph with eosinophil counts in several high-power fields to compare them between groups.
MINNOR COMMENTS
- Page 3, Line 111: Did you use reducing or non-reducing conditions to perform SDS-Page? Please, specify.
- Page 4, Lines 151-152: Please, specify the conjugated-fluorochrome for each antibody and the manufacturer. Also “Singlec-F” is wrong; the correct name is “Siglec-F” (also this name appears in axis from Figure 6). Did you add Fc block before or simultaneously to adding the antibody mix? What isotype controls did you use?
- Page 4, Line 175: What post-hoc test to correct p value did you use in ANOVA analysis (Bonferroni, Dunn, etc.)?
- Page 5, Line 190: Is it possible to improve image quality to observe better the dimeric protein? Is it possible that rAlt a 1 show a molecular weight of 25kDa because the SDS-page was not done in reducing conditions?
- Page 5, Line 198: “No statistically differences (p>0.01)…” In methods sections, you established as significant a p-value less than 0.05. Please, clarify.
- Page 5, Line 202: “** P<0.01” is not shown in the graph. Add or remove it.
- Page 6, Line 212: Is it possible to measure TSLP levels in lung?
- Page 10, Lines 345-374: “Terrier et al. (2010) demonstrated that the IL-17E can release IL-5 and provoke the recruitment of eosinophils”. This phrase is confused. Please reformulate it (IL-17E do not release IL-5, IL-17E inducing IL-5 releasing through other cell types).
Author Response
MAJOR COMMENTS
Figure 2, 3 and 4: Comparisons between groups (Extract, nAlt a 1 and rAlt a 1) are missed. Are there statistical differences between these groups?
Thank you for your comment.
The statistical differences between all groups have been included in figures 2, 3 and 4 as horizontal bars with the corresponding signification (ns; *; **; ***).
The results have been explained in the text lines 205-212:
Quantification of total IgE in mouse serum
Figure 2 shows that all groups instilled with A. alternata extract or Alt a 1 (native and recombinant) achieved IgE levels with statistically significant differences (p<0.001) compared to the mice administered with PBS. The highest IgE levels were obtained by the group of mice instilled with nAlt a 1 compared to control groups or with rAlt a 1 instilled mice (p<0.001). No statistically significant differences (p>0.05) were demonstrated between the groups instilled with A. alternata extract and the group of mice instilled with rAlt a 1.
Lines 218-227:
Cytokine quantification in lung homogenate
Instillation of nAlt a 1 resulted in an increase in IL-5, IL-13, IL17E and IL-33 in lung homogenates compared to the PBS control group (p<0.001) (Figure 3). Instillation of rAlt a 1 induced the IL-5, IL-13 and IL-33 cytokines production (p<0.001) but, the IL-17E levels induced by rAlt a 1 demonstrated no statistically significant differences (p>0.05) compared with the control groups. An increase of IL-17E and IL-5 could be observed in the groups of animals instilled with A. alternata extract compared to the PBS instilled mice control group (p<0.001). The IL-13 and IL-33 cytokine levels induced by nAlt a 1 and rAlt a 1 instillation were higher than those induced by A. alternata extract (p<0.001). No statistically significant differences (p>0.05) in IL-13 and IL-33 levels were demonstrated between A. alternata extract and PBS instilled mice.
Lines 234-248
Cytokine quantification in spleen homogenate
Figure 4 shows that instillation of nAlt a 1 induced the IL-5, IL-17E and IL-33 cytokines production (p<0.001) but, no statistically significant differences (p>0.05) were demonstrated in IL-13 levels produced by nAlt a 1 instilled mice compared to the PBS control group. Instillation of rAlt a 1 resulted in an increase in IL-5, IL-13, IL-17E and IL-33 levels in spleen homogenates compared to the PBS control group (p<0.05 and p<0.001). The IL-13 and IL-17E levels were increased (p<0.001) in mice instilled with A. alternata extract compared to the PBS control group but, no statistically significant differences (p>0.05) were observed in IL-17E levels between the treated groups and the mice instilled with A. alternata extract. No statistically significant differences (p>0.05) were demonstrated in the production of IL-5 and IL-33 cytokines between the control groups. The IL-33 cytokine levels induced by nAlt a 1 or rAlt a 1 treated groups were higher than those induced by A. alternata extract (p<0.001). This control group demonstrated no statistically significant differences (p>0.05) compared to the PBS instilled mice.
Page 7, Lines 236-237: Why did you not measure lymphocyte proliferation in mice instilled with rAlt a 1?
Lymphocyte proliferation could not be measured in mice instilled with rAlt a 1 because the batch of the recombinant protein used in the study was exhausted at the time of the assay. We are re-running this experiment, but it may be some time before we can produce the recombinant protein again. Nevertheless, we think the results obtained with nAlt a 1 provide information on the experimental model induced by the molecule. Based on the results obtained, it is expected that the recombinant protein will behave similarly to the native protein in this assay.
Page 6 and 7, Lines 243-255 (Flow cytometry results): In my opinion I believe that these experiments are not well conducted. First, you must show the gating strategy used (FSC vs SSC, single cells, CD45+ cells, SIGLEC-F+/CD11B+ cells, etc.). Second, in dot plots I do not distinguish independent populations and fluorescence intensity are very low; please, show me isotype controls that you use, and the gating strategy followed. How do you calculate the number eosinophils/mL, what software did you use? I think that the better data is eosinophil percentage. Also, you must add a graph where you show the percentage values of eosinophils from each mouse, and reflect on it the significant differences.
The strategy we followed to test for the presence of eosinophils was as follows. First, the cells were selected with the CD45 antibody, which is the useful marker for leukocytes. Positive cells were then challenged with CD11c antibody. All cells that were positive for CD45+ and negative for CD11c- were selected and challenged with the Siglec-F and CD11b antibody. Those cells that were Siglec-F+ and CD11b+ were considered as eosinophils (according to Stevens et al.).
A vial of cells without antibody was used as a control. We understand that isotype controls can be a valuable tool but their use divides researchers (Herzenberg et al. 2006 and Maecker et al. 2006). In our case, we did not use an isotype control, because we considered that the expected cell expression levels were high and, therefore, the antibodies would bind only to the desired cells and not to artefacts.
We received a report from the laboratory specifying both the number of eosinophils and the percentage detected. All results were analyzed with the flow cytometer software itself but, we made a mistake in writing the results, they are not eosinophils/ml, but percentage of eosinophils.
Following your suggestion, we have modified the results section and replaced the flow cytometry graphs with a graph showing the average percentage of eosinophils. In order not to have many graphics, we have grouped the number of eosinophils detected in each mouse and the comparison between the different groups into a single graph. We believe this change will make the results clearer and more understandable.
Page 8, Line 262 (Figure 7): It would be interesting to include a graph with eosinophil counts in several high-power fields to compare them between groups.
Thank you for your suggestion. Instead of the eosinophil count in various high power fields we have included a graph (Figure 6) with the percentage of eosinophils counted by flow cytometry in each group studied along with the statistical differences between them.
MINNOR COMMENTS
Page 4, Lines 151-152: Please, specify the conjugated-fluorochrome for each antibody and the manufacturer. Also “Singlec-F” is wrong; the correct name is “Siglec-F” (also this name appears in axis from Figure 6). Did you add Fc block before or simultaneously to adding the antibody mix? What isotype controls did you use?
We apologise for having misspelled the word Siglec-F. The conjugated-fluorochrome for each antibody are as follow: CD45 (Alexa Fluor 488), Siglec-F (PE), CD11c (Efluor 450) and CD11b (APC). The Fc block was added to the cells 30 min before adding the antibody mix. As advised by the Analytical and High Resolution Microscopy Service (SGIKER) of the University of the Basque Country, only a vial of cells without antibody was used as a control.
Page 4, Line 175: What post-hoc test to correct p value did you use in ANOVA analysis (Bonferroni, Dunn, etc.)?
We have analysed the results with the post-hoc Bonferroni test.
Page 3, Line 111: Did you use reducing or non-reducing conditions to perform SDS-Page? Please, specify and Page 5, Line 190: Is it possible to improve image quality to observe better the dimeric protein? Is it possible that rAlt a 1 show a molecular weight of 25kDa because the SDS-page was not done in reducing conditions?
All SDS-PAGE analyses were performed under reducing conditions. The molecular weight of the recombinant protein was 25 kDa, and it did not show the dimeric form. This could be due to the recombinant nature of the protein produced in the prokaryotic E. coli system. As the molecular weight was not as expected, the purified rAlt a 1 was sent for sequencing analysis to confirm that it was the correct protein. Mass spectrometry results showed that the recombinant Alt a 1 produced in our laboratory was 98% homologous to native Alt a 1 (NCBI. BBK07919.1)
Page 5, Line 198: “No statistically differences (p>0.01)…” In methods sections, you established as significant a p-value less than 0.05. Please, clarify.
This is a mistake that it is already corrected in the manuscript. In line 210 it should appear now "no statiscally differences (p>0.05)" as the methods section states.
Page 5, Line 202: “** P<0.01” is not shown in the graph. Add or remove it.
Thank you for your suggestion, we have remove it.
Page 6, Line 212: Is it possible to measure TSLP levels in lung?
We know that TSLP is an important cytokine in allergy, but we decided to measure the cytokines IL-25 and IL-33 because these “alarmin” cytokines can induce both innate and adaptive immune responses and shift inflammation towards a type 2 immune response such as TSLP. We felt that they provided sufficient information about allergic symptomatology in the murine model.
Now, we do not have enough samples to be able to repeat the analyses with TSLP, but we will include this cytokine in future experiments.
Page 10, Lines 345-374: “Terrier et al. (2010) demonstrated that the IL-17E can release IL-5 and provoke the recruitment of eosinophils”. This phrase is confused. Please reformulate it (IL-17E do not release IL-5, IL-17E inducing IL-5 releasing through other cell types).
We have rephrased it to: “Terrier et al. demonstrated that IL-17E can induce the release of IL-5 through mononuclear cells and cause eosinophil recruitment.”
Reviewer 2 Report
It is an interesting and well-written article. Some changes are needed, mainly in the results presentation.
If you use Leino et al. (2013) and Causton et al. (2018) and similar sentences like that, it is not necessary that you mention the year
“With this in mind” is not very scientific. You should change or omit this part of the sentence
In the results section you should only mention your results. The comparison with other studies or clinical trials can be described in the discussion. The results 186 obtained from mass spectrometry of the rAlt a 1 protein agree with those published in 187 NCBI (BBK07919.1) and sequencing analysis of rAlt a 1 demonstrated a homology of 98 % 188 with the native Alt a 1 sequence. This sentence should be written in the discussion
Line 198 and 209. In the material and methods you said that p<0.05 is considered statically significative, as usually. Nevertheless in this line you say No statistically significant differences (p>0.01), could you say what is the real p value?
Figures 2-5. Please explain the statistically analysis better in the legends. It is not clear what groups you are comparing. I suppose you use one-way ANOVA post-hoc Bonerroni correction? You should specify it and explain the groups that you are comparing.
I recommend also that you do a table with the absolute values of all the parameters in all groups and represent the p value of one-way ANOVA and p values after each comparison post-hoc Bonerroni correction
Author Response
If you use Leino et al. (2013) and Causton et al. (2018) and similar sentences like that, it is not necessary that you mention the year
Thank you very much for your comment, we have changed it.
“With this in mind” is not very scientific. You should change or omit this part of the sentence
We have changed the phrase as you suggested and omit “with this in mind.”
In the results section you should only mention your results. The comparison with other studies or clinical trials can be described in the discussion. The results 186 obtained from mass spectrometry of the rAlt a 1 protein agree with those published in 187 NCBI (BBK07919.1) and sequencing analysis of rAlt a 1 demonstrated a homology of 98 % 188 with the native Alt a 1 sequence. This sentence should be written in the discussion
We have modified the phrase to read as follows: “The results obtained by mass spectrometry of the rAlt a 1 protein showed 98% homology with the native Alt a 1 sequence (NCBI. BBK07919.1).”
Line 198 and 209. In the material and methods you said that p<0.05 is considered statically significative, as usually. Nevertheless in this line you say No statistically significant differences (p>0.01), could you say what is the real p value?
This is a mistake that it is already corrected in the manuscript. In line 210 it should appear now "no statiscally differences (p>0.05)" as the methods section states.
Figures 2-5. Please explain the statistically analysis better in the legends. It is not clear what groups you are comparing. I suppose you use one-way ANOVA post-hoc Bonerroni correction? You should specify it and explain the groups that you are comparing.
Statistical analysis was performed with Prism software (version 7.0, GraphPad Software). All the results were analyzed through one-way ANOVA and Bonferroni post-hoc tests. Statistical differences between all groups have been better explained in the legends and in the text itself.
I recommend also that you do a table with the absolute values of all the parameters in all groups and represent the p value of one-way ANOVA and p values after each comparison post-hoc Bonferroni correction
Thank you for your input. Statistical differences between all groups have been included in the graphs and are also explained in the text.
Reviewer 3 Report
The subject of this research is Alt a 1 which, based on previous in vitro studies, is a potential risk factor for severe asthma. In Introduction the authors in sufficiently covered the subject of asthma, fungi and their role in allergies and exacerbation of the disease. Although I highly support publishing of the manuscript there are several things that should be corrected:
Please address the following:
- why 20 μg of A. alternata extract was used while purified protein, extracted and recombinant, were applied in 5 μg? Have you assessed the amount of Alt a 1 in the extract?
- Why lymphocyte proliferation was not done for recombinant protein (figure 5)?
- Figure 6 is not clearly explained. What are 1F-F4?
- Please improve the quality of Figure 7
- Part from line 269-294 should better fit in the introduction part of the article. Please replace it.
- The sentence in line 366-370 should be corrected, it sounds confusing
Author Response
Why 20 μg of A. alternata extract was used while purified protein, extracted and recombinant, were applied in 5 μg? Have you assessed the amount of Alt a 1 in the extract?
The amount of Alt a 1 present in the extract was 10 times lower than nAlt a 1 or rAlt a 1 (measured by ELISA). In our previous studies (data no published) 20 μg of extract and 5 μg of purified proteins resulted in the minimum quantity able to induce a significate immune response in mice compared with the PBS control group. Taking into account that the fungal extract possesses other proteins and allergens that can act as adjuvants, this quantity let us to minimize the effect of this other proteins in the A. alternata instilled control group mice and, that the mice instilled a secure quantity of extract through their snout.
There is no consensus among researchers on the amount of A. alternata extract to instill in mice to trigger asthma. The amounts used range from 20 ug (Oczypok et al. 2015; Löser et al. 2017) to 100 ug (Doherty et al. 2012 and 2013). Based on our previous studies and in the study of Havaux et al. (2005), we selected to use 20 μg of fungal extract to achieve the asthmatic response.
Oczypok EA, Milutinovic PS, Alcorn JF, Khare A, Crum LT, Manni ML, Epperly MW, Pawluk AM, Ray A, Oury TD. Pulmonary receptor for advanced glycation end-products promotes asthma pathogenesis through IL-33 and accumulation of group 2 innate lymphoid cells. J Allergy Clin Immunol. 2015 Sep;136(3):747-756.e4. doi: 10.1016/j.jaci.2015.03.011. Epub 2015 Apr 28. PMID: 25930197; PMCID: PMC4562894.
Löser S, Gregory LG, Zhang Y, Schaefer K, Walker SA, Buckley J, Denney L, Dean CH, Cookson WOC, Moffatt MF, Lloyd CM. Pulmonary ORMDL3 is critical for induction of Alternaria-induced allergic airways disease. J Allergy Clin Immunol. 2017 May;139(5):1496-1507.e3. doi: 10.1016/j.jaci.2016.07.033. Epub 2016 Sep 10. PMID: 27623174; PMCID: PMC5415707.
Havaux X, Zeine A, Dits A, Denis O. A new mouse model of lung allergy induced by the spores of Alternaria alternata and Cladosporium herbarum molds. Clin Exp Immunol. 2005 Feb;139(2):179-88. doi: 10.1111/j.1365-2249.2004.02679.x. PMID: 15654816; PMCID: PMC1809297.
Doherty TA, Khorram N, Sugimoto K, Sheppard D, Rosenthal P, Cho JY, Pham A, Miller M, Croft M, Broide DH. Alternaria induces STAT6-dependent acute airway eosinophilia and epithelial FIZZ1 expression that promotes airway fibrosis and epithelial thickness. J Immunol. 2012 Mar 15;188(6):2622-9. doi: 10.4049/jimmunol.1101632. Epub 2012 Feb 10. PMID: 22327070; PMCID: PMC3294141.
Doherty TA, Khorram N, Lund S, Mehta AK, Croft M, Broide DH. Lung type 2 innate lymphoid cells express cysteinyl leukotriene receptor 1, which regulates TH2 cytokine production. J Allergy Clin Immunol. 2013 Jul;132(1):205-13. doi: 10.1016/j.jaci.2013.03.048. Epub 2013 May 17. PMID: 23688412; PMCID: PMC3704056.
Why lymphocyte proliferation was not done for recombinant protein (figure 5)?
Lymphocyte proliferation could not be measured in mice instilled with rAlt a 1 because the batch of the recombinant protein used in the study was exhausted at the time of the assay. We are re-running this experiment, but it may be some time before we can produce the recombinant protein again. Nevertheless, we think the results obtained with nAlt a 1 provide information on the experimental model induced by the molecule. Based on the results obtained, it is expected that the recombinant protein will behave similarly to the native protein in this assay.
Figure 6 is not clearly explained. What are 1F-F4?
The strategy we followed to test for the presence of eosinophils was as follows. First, the cells were selected with the CD45 antibody, which is a useful marker for leukocytes. Positive cells were then challenged with CD11c antibody. All cells that were positive for CD45+ and negative to CD11c- were selected (region F) and challenged with the Siglec-F and CD11b antibody. Those cells that were Siglec-F+ and CD11b+ positive were considered as eosinophils (region F4). The other regions, F1, F2 and F3, do not meet the Siglec-F+ and CD11b+ positive criteria and therefore cannot be considered as eosinophils. However, we have replaced this graph by a clearer one (eosinophil count in percentage form), where statistically significant differences between the different groups can be observed (Figure 6).
Please improve the quality of Figure 7
Thank you for your comment. We have improve the image as much as we were able to.
Part from line 269-294 should better fit in the introduction part of the article. Please replace it.
Thank you for your comment. We have move almost all this paragraph to the introduction to lines 45-53. The references have been re-arranged according to the change.
The sentence in line 366-370 should be corrected, it sounds confusing
We have corrected the sentence and it is now as follows: “The establishment of this allergic asthma model induced by Alt a 1, the main responsible of A. alternata fungal asthma and its exacerbation [36,43], will allow the evaluation of the immune-regulatory or immune-tolerant
Round 2
Reviewer 1 Report
Thank you for your accurate responses. Authors have solved almost all my previous comments. I only have a few minor comments:
- Page 6, Lines 283-285: “An increase of IL-17E and IL-5 could be observed in the groups of animals instilled with A. alternata extract compared to the PBS instilled mice (p<0.001)”. The differences between control group and mice instilled with A. alternata extract are p < 0.05 for IL-5 and p < 0.01 for IL-17E, not p<0.001 as you show in the text.
- To be consistent with the format used in the whole article, Figure 6 should be drawn as the others (representing dots and not bars). In addition, on the y-axis it should read "Eosinophils (%)", to homogenize the format with the rest of the figures.
Author Response
Page 6, Lines 283-285: “An increase of IL-17E and IL-5 could be observed in the groups of animals instilled with A. alternata extract compared to the PBS instilled mice (p<0.001)”. The differences between control group and mice instilled with A. alternata extract are p < 0.05 for IL-5 and p < 0.01 for IL-17E, not p<0.001 as you show in the text.
Thank you for your comment. We have corrected it and the sentence is now as follows:” An increase in IL-17E and IL-5 was observed in the groups of animals instilled with A. alternata extract compared to mice instilled with PBS (p<0.01 and p<0.05, respectively).”
To be consistent with the format used in the whole article, Figure 6 should be drawn as the others (representing dots and not bars). In addition, on the y-axis it should read "Eosinophils (%)", to homogenize the format with the rest of the figures.
Thank you for your feedback. We have changed the bar graph in Figure 6 to a dot graph to make it homogeneous with the other figures and added "Eosinophils (%)" to the y-axis.